# Moderating Effect of Muscular Strength in the Association Between Cancer and Depressive Symptomatology

**DOI:** 10.3390/healthcare13070715

**Published:** 2025-03-24

**Authors:** Diogo Veiga, Marcelo de Maio Nascimento, Miguel Peralta, Élvio R. Gouveia, Adilson Marques

**Affiliations:** 1Centro Interdisciplinar de Performance Humana, (CIPER) Faculdade de Motricidade Humana, Universidade de Lisboa, 1499-002 Cruz-Quebrada, Portugal; dmcveiga@fmh.ulisboa.pt (D.V.); mperalta@fmh.ulisboa.pt (M.P.); 2Department of Physical Education, Federal University of Vale do São Francisco, Petrolina 56304-205, Brazil; marcelo.nascimento@univasf.edu.br; 3Swiss Center of Expertise in Life Course Research LIVES, 1201 Geneva, Switzerland; 4Instituto de Saúde Ambiental (ISAMB), Faculdade de Medicina, Universidade de Lisboa, 1649-026 Lisboa, Portugal; 5Department of Physical Education and Sport, University of Madeira, 9020-105 Funchal, Portugal; erubiog@staff.uma.pt; 6Laboratory for Robotics and Engineering Systems (LARSYS), Interactive Technologies Institute, 9020-105 Funchal, Portugal

**Keywords:** depression, elderly, fitness, handgrip, moderation, preventive medicine

## Abstract

Background/Objectives: Depression, as one of the leading causes of disease burden, frequently co-occurs with other diseases. Cancer seems to be strongly associated with depression more than any other disease. As an outcome of physical fitness, muscular strength seems to have a protective effect on depression. This study aimed to analyze how muscular strength moderates the relationship between cancer and depressive symptomatology among older European adults. Methods: Cross-sectional data from wave 8 (2019/2020), including 41,666 participants (17,986 males) of the population-based Survey of Health, Aging, and Retirement in Europe, were analyzed. Grip strength, used as the moderator, was measured twice on each hand using a dynamometer. The EURO-D 12-item scale was used to measure depressive symptomatology. Results: Grip strength had a significant effect as a moderator in the association between cancer and depressive symptoms (male: B = −0.025, 95% CI = −0.04, −0.01; female: B = −0.02, 95% CI = −0.04, 0.00). Also, the grip strength moderation values are below 55.3 kg for males and 39.4 kg for females. Conclusions: Muscular strength, as measured by grip strength, moderated the relationship between cancer and depressive symptomatology. This supports the theory that recovery programs could include physical activity, namely muscle-strengthening exercises, to prevent depression.

## 1. Introduction

Global statistics indicate that, in 2022, there were approximately 20 million new cases of cancer, followed by 10 million deaths [1]. As a result of population aging and expansion, it is estimated that, each year until 2040, cases of cancer will increase by approximately 30 million [2]. For these reasons, in the 21st century, cancer has been considered a major public health and economic problem [1]. On the other hand, due to innovative treatments, the number of cancer survivors has also been increasing [3]. Cancer survivor is defined as an individual diagnosed with some type of cancer, including before, during, or after treatment [4].

However, survivors must learn to cope with a range of side effects and long-term consequences of chemotherapy treatment [5]. Due to gastrointestinal changes, it is common for survivors to experience nausea and vomiting [6]. Nevertheless, the side effects of chemotherapy still include the malabsorption of food, anorexia, diarrhea, dehydration and electrolyte imbalance, weight loss, anemia, cognitive deficits, sleep disorders, and fatigue [6,7]. Consequently, the combination of changes in nutritional status and inflammatory processes favors cancer cachexia (a complex syndrome characterized by severe weight loss, muscle wasting, and metabolic alterations), significantly impacting muscle strength and overall physical function [8,9].

Due to all this, psychological conditions such as anxiety and depression are common in cancer survivors [10]. Prevalence rates for depression vary greatly depending on criteria and evaluation techniques. As stated by the World Health Organization [11], around 280 million people worldwide, or approximately 5% of the adult population, suffer from some form of depressive symptoms. Thus, depressive symptoms are also considered a serious common global health problem. Research has shown that the prevalence of major depression and broader depression spectrum disorders in cancer patients ranges from 0% to 38% and from 0% to 58%, respectively [12]. The co-occurrence of cancer with depressive symptoms has been attributed to the series of physiological and functional changes that abruptly hinder or interrupt the performance of activities of daily living [13]. Therefore, the combination of cancer and depression increases the risk of non-adherence to medical treatment (reduced survival time) or being the trigger for suicide [14].

In recent years, studies have documented the negative association between cancer and depressive symptoms for muscle strength. Grip strength is a valid and reliable indicator of upper body strength and a proxy for total muscular strength that is correlated with levels of physical activity [15]. A prospective cohort study that included 445,552 individuals from the UK Biobank found that grip strength was inversely associated with eight cancer sites and cancer from all causes [16]. In turn, a current study revealed that cancer survivors with low handgrip strength showed a 2.02-fold increased risk for depression [17]. Although the evidence presented has shown a strong association between cancer, depressive symptoms, and muscular strength, to date, no study has evaluated in a large sample of Europeans, specifically during aging, the role of muscular strength in the relationship between cancer and depression. Thus, this study aimed to analyze how muscular strength moderates the relationship between cancer and depressive symptoms among middle-aged and older European adults.

## 2. Materials and Methods

### 2.1. Participants and Procedures

This study was based on data from wave 8 (2019/2020) of the population-based Survey of Health, Aging, and Retirement in Europe (SHARE). The SHARE methodology has been previously described [18]. Every two years, since 2002, SHARE gathers data on people in many European nations as well as Israel who are 50 years of age or older. The SHARE data collection procedures are subject to continuous ethics review. For more details, please see: https://share-eric.eu/fileadmin/user_upload/Ethics_Documentation/SHARE_ethics_approvals.pdf (accessed on 21 September 2024). Participants were excluded if they did not live at the sampled address (for example, because it was a vacation or seasonal residence), were physically or psychologically unable to participate, had passed away before to the start of the field session, or could not speak the national questionnaire’s specified language. The protocols to ensure data privacy and confidentiality were verified by the Ethics Council of the Max-Planck Society for the Advancement of Science and the University of Mannheim, who accepted the SHARE protocol. Written informed permission was acquired from every study participant.

The final sample comprised 41,666 participants (17,986 males and 23,680 females), mean age of 70.65 (9.1) years, from 29 countries (Austria, Germany, Sweden, Netherlands, Spain, Italy, France, Denmark, Greece, Switzerland, Belgium, Israel, Czech Republic, Poland, Ireland, Luxembourg, Hungary, Portugal, Slovenia, Estonia, Croatia, Lithuania, Bulgaria, Cyprus, Finland, Latvia, Malta, Romania, and Slovakia). SHARE data collection is based on computer-assisted personal interviewing (CAPI). The interviewers conducted face-to-face interviews using a laptop on which the CAPI instrument was installed. The data are freely accessible. The data can be accessed through the SHARE project website: https://share-eric.eu/ (accessed on 21 January 2023).

### 2.2. Measures

Depressive symptomatology was used as the outcome measure. The EURO-D 12-item scale was used to measure depressive symptomatology. The score ranged between 0 and 12, with higher scores indicative of higher depressive symptoms. A cut-off point of ≥4 points diagnosed a clinically significant depression [19,20]. The scale description and validation are described elsewhere [20].

The exposure measure was previously used with cancer. Participants were asked to report the presence or absence of cancer previously diagnosed by a medical doctor. The participants had to answer the question: “*Has a doctor ever told you that you had/ Do you currently have any of the conditions on this card? (With this, we mean that a doctor has told you that you have this condition and that you are either currently being treated for or bothered by this condition)*”.

Grip strength was used as the moderator. It was measured twice on each hand using a dynamometer (Smedley, S Dynamometer, TTM, Tokyo, 100 kg) [21]. Participants were standing or sitting, with the elbow at a 90° angle, the wrist in a neutral position, keeping the upper arm tight against the trunk, and the inner lever of the dynamometer adjusted to the hand. Participants squeezed the dynamometer with their hands as hard as possible and sustained it for 5 seconds. Before the assessment, participants had the opportunity to train. Values were recorded twice for each hand, alternating between the left and right. The grip strength variable contained the maximum value (the left or the right hand) of the grip strength measurement of both hands. Valid measurements were values of two in one hand that differed by less than 20 kg. Grip strength measurements with values equal to 0 kg or higher than 100 kg were excluded, as well as if grip strength was only measured once in one hand.

### 2.3. Covariates

Three self-reported variables were considered as confounding factors: age, country, and physical activity. Physical activity was measured as “frequency of moderate physical activity” (e.g., gardening, cleaning the car, walking) and “frequency of vigorous physical activity” (e.g., sports, heavy housework, a job that involves physical labor). The response alternatives for moderate and vigorous activity were: (1) more than once a week, (2) once a week, (3) up to three times a month, and (4) almost never or never. The last two response options were grouped into a category called less than once a week, according to a previous study.

### 2.4. Statistical Analysis

Descriptive statistics were calculated for all variables (means, standard deviation, and frequencies) for the entire sample and stratified by sex. The comparison of depressive symptomatology between males and females according to cancer diagnosis was tested by an independent sample t-test. The correlation between grip strength and depressive symptomatology was assessed by a Pearson correlation analysis [22]. The moderation analysis of grip strength (moderator, W) on the relationship between cancer (categorical, X) and depressive symptomatology (continuous, Y) was based on the moderation paths proposed by Baron and Kenny [23], as shown in Figure 1. Andrew Hayes’ PROCESS macro-3.5 was used to conduct the moderation analysis stratified by sex. For this simple moderation model, the process macro automatically centers the variables, computes the interaction term, runs the regression model with the interaction term, and then tests the simple slopes. The Johnson–Neyman approach was applied to test statistically significant interactions and identify regions of significance. This procedure was also used to obtain a threshold of statistical significance. Analyses were stratified by sex and controlled for a series of covariates (e.g., physical activity, age, country). Data analysis was performed using IBM SPSS Statistics version 28 (SPSS Inc., an IBM Company, Chicago, IL, USA). For all tests, the statistical significance was set at *p* < 0.05.

## 3. Results

Descriptive analyses of the total sample and according to sex are presented in Table 1. A greater percentage of females (30.5%) reported depressive symptoms than males (18.3%). On the other hand, more males (5.7%) reported being diagnosed with cancer than females (4.6%). 

Table 2 presents he correlation between grip strength and depressive symptoms. Grip strength was significantly and negatively related to depressive symptoms for the total sample (*r* = −0.254, *p* < 0.001), for males (*r* = −0.193, p < 0.001), and females (*r* = −0.210, *p* < 0.001).

Table 3 compares depressive symptoms between participants diagnosed and not diagnosed with cancer in the total sample and according to sex. The mean depressive symptomatology was higher in participants diagnosed with cancer, regardless of sex (male: 2.66 vs. 1.86, *p* < 0.001; female: 3.39 vs. 2.59, *p* < 0.001). Regardless of the presence or absence of a cancer diagnosis, females consistently reported higher average scores of depressive symptoms than males.

Table 4 shows the moderation effect of grip strength (W) on the relationship between cancer (X) and depressive symptomatology (Y) for males and females. Grip strength was a significant moderator in the association between cancer and depressive symptoms (male: B = −0.03, 95% CI = −0.04, −0.01; female: B = −0.02, 95% CI = −0.04, 0.00). Also, according to the Johnson–Neyman test, the grip strength moderation values in the significance region were below 55.3 kg for males and 39.4 kg for females.

## 4. Discussion

This study aimed to analyze how muscular strength moderates the relationship between cancer and depressive symptomatology in European middle-aged and older adults. The findings show that muscular strength was significantly associated with fewer depressive symptoms in middle-aged and older people and that both male and female with cancer had more depressive symptoms. Furthermore, when facing a cancer diagnosis, muscular strength was a moderator of depressive symptomatology, attenuating its association with cancer.

In line with previous studies, depressive symptoms were more prevalent in females compared to males [24,25]. Differences in depression rates between the sexes have been attributed to a number of biological (e.g., hormonal fluctuations) [26], psychological (e.g., socialization and coping mechanisms) [27], and genetic (e.g., family history) factors [28]. Analyses revealed an inverse correlation between grip strength and depressive symptoms, suggesting that a decline in grip strength over time is associated with an increased risk of developing depression [29]. The inverse correlation between grip strength and depressive symptoms, suggesting that a decline in grip strength over time is associated with an increased risk of developing depression [30]. Grip strength has been proposed as a broadly consistent biomarker used to assess and explain musculoskeletal function (i.e., functional capacity) in several populations [31], including depression [32].

Our findings reinforce that persons diagnosed with cancer present more depressive symptoms. A study that included patients hospitalized with lung cancer in general hospitals revealed that 43.2% presented clinically significant depressive symptoms [33]. In turn, among cancer patients receiving medical care at a hospital in Serbia, mild depressive symptoms were identified in 27.2%, moderate in 22%, and severe in 18% [34], while the severity of depression was greater in middle-aged and older adults. The facts highlight the importance of examining depressive symptoms in cancer survivors, especially through the use of simple, low-cost measures, such as grip strength, as this association appears to be present regardless of sex and type of cancer [35,36].

Our primary analysis suggested that grip strength played a moderating role in the onset of depressive symptoms in cancer survivors. Although this effect was small, it was significant. Based on the findings, from a clinical perspective, it is not possible to state that grip strength, more specifically the measurement of muscle strength, is sufficient to treat depression in cancer survivors. However, the finding may serve as a strategy capable of alerting and assisting professionals about the presence of depressive symptoms in the presence of values below 55.3 kg for males and 39.4 kg for females. However, this would not be expected, since even pharmacological therapies for depression show significant effects in only about 30% of cancer cases, and it is possible that patients suffer a relapse if the medication is stopped [37]. In turn, there is no convincing evidence that one antidepressant is more effective than another, according to the Food and Drug Administration [38]. On the other hand, it is known that the combination of medication and physical exercise (i.e. strength training and aerobic endurance) can bring significant gains for cancer survivors [39]. Thus, exercise should not be considered as a monotherapy, but rather as an adjuvant therapy in the treatment of cancer and depressive symptoms [40].

It is important to highlight that many other variables may have influenced the relationship between cancer and depressive symptoms in the population evaluated. According to the literature, this relationship is expected to be influenced by nutrition [41], smoking habits [42], and alcohol consumption [43]. Therefore, the observed moderating effect suggested that therapies focused on physical strength and functionality may be beneficial strategies for reducing mental changes linked to cancer.

### Limitations

Being the first study to analyze the moderating effect of muscle strength on the association between cancer and depression, including the middle-aged and older adult population of 29 European countries, is one of the strengths of the present study. Secondly, the analyses were differentiated according to sex. Thirdly, the use of the Johnson–Neyman test allowed us to identify significant threshold values of moderation of muscle strength on the relationship between depressive symptoms and cancer for both sexes. On the other hand, the study has limitations. First, grip strength measures that differed by less than 20 kg were considered valid. A 20 kg variation seems substantial. In fact, it is nearly equal to the female stratum mean. Though grip strength varies by population and geography, there may be a significant intra-individual variation if measures range by as much as 20 kg. Regardless, this is the assessment of the data made by the SHARE study team. Second, due to the correlational cross-sectional design, causal associations were limited. Third, physical activity was measured by self-reported questionnaires, which may have generated higher estimates than when assessed by objective measures. Other limitations of the study are: First, due to the correlational cross-sectional design, causal associations were limited. Second, physical activity was measured by self-reported questionnaires, which may have generated higher estimates than when assessed by objective measures [44]. On the other hand, the inclusion of a model with grip strength as a moderator measured by an objective physical fitness test may have compensated for this limitation. Third, the SHARE database does not provide for a series of information, such as nutrition, smoking, alcohol use/abuse, history of depression, medications, and type of cancer. Therefore, it was not possible to control the analyses for a series of important confounders. 

In turn, due to the low value of the moderation effect found, we recognize that further research is needed. Thus, it is suggested that future studies control the analyses for a series of confounding factors associated with cancer, depression, and muscle strength. Furthermore, investigations can replicate our analyses comparing data by age groups, years of education, income, and different regions of Europe. It is likely that physical limitations associated with reduced grip strength have caused an increase in the vulnerability of cancer survivors, potentiating depressive symptoms [45]. Thus, it is suggested that future analyses on the relationship between cancer and depressive symptoms include a categorical moderator composed of three levels of physical activity (i.e., low, medium, high). In addition, it would be interesting to adopt longitudinal designs to confirm potential causal relationships and explore the underlying mechanisms that may explain the moderating role of muscle strength. Finally, it is suggested that middle-aged and especially older adult individuals regularly participate in exercise programs that strengthen muscle strength and general physical condition, since, as evidenced in this study, a higher grip strength can serve as a buffer, affecting coping strategies and resilience in cancer survivors diagnosed with depression.

## 5. Conclusions

Comparatively, depressive symptoms were prevalent in females. On the other hand, cancer diagnosis was more frequent among males. Inverse correlations between grip strength and depressive symptoms were identified, suggesting that a decline in grip strength over time is associated with an increased risk of developing depression. Muscular strength, as measured by grip strength, acted as a moderator of depressive symptomatology, attenuating its association with cancer. This supports the theory that recovery programs could include physical activity, namely muscle-strengthening exercises, to prevent depression.

## Figures and Tables

**Figure 1 healthcare-13-00715-f001:**
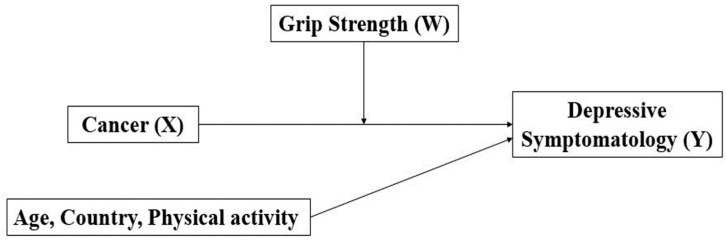
A conceptual diagram of the relationship between cancer (X) and depressive symptomatology (Y) moderated by grip strength (W).

**Table 1 healthcare-13-00715-t001:** Sample characteristics for the total sample and by sex.

	Mean (SD) or *n* (%)			
	Total (*n* = 41,666)	Male (*n* = 17,986)	Female (*n* = 23,680)	*p*
Age (years)	70.65 (9.14)	71.12 (8.77)	70.29 (9.40)	<0.001
Grip strength (kg)	32.04 (11.18)	40.68 (9.93)	25.47 (6.72)	<0.001
EURO-D score	2.32 (2.16)	1.90 (1.95)	2.63 (2.25)	<0.001
EURO-D ≥ 4 Yes [*n* (%)] No [*n* (%)]	10,523 (25.3) 31,143 (74.7)	3295 (18.3) 14,691 (81.7)	7228 (30.5) 16,452 (69.5)	<0.001
MPA [*n* (%)] <1x/week 1/week >1/week	7696 (18.5) 6215 (14.9) 27,755 (66.6)	3166 (17.6) 2695 (15.0) 12,125 (67.4)	4530 (19.1) 3520 (14.9) 15,630 (66.0)	<0.001
VPA [*n* (%)] <1x/week 1/week >1/week	21,788 (52.3) 6326 (15.2) 13,552 (32.5)	8719 (48.5) 2699 (14.5, 15.0) 6568 (36.5)	13,069 (55.2) 3627 (15.3) 6984 (29.5)	<0.001
Cancer Yes [*n* (%)] No [*n* (%)]	2117 (5.1) 39,549 (94.9)	1032 (5.7) 16,954 (94.3)	1085 (4.6) 22,595 (95.4)	<0.001

Abbreviations: SD, standard deviation.

**Table 2 healthcare-13-00715-t002:** Pearson correlation between grip strength and depressive symptomatology.

	Depressive Symptoms (EURO-D 12 Score)					
	Total Sample		Male		Female	
	*r*	*p*	*r*	*p*	*r*	*p*
Grip strength	−0.254	<0.001	−0.193	<0.001	−0.210	<0.001

**Table 3 healthcare-13-00715-t003:** Comparison of depressive symptomatology according to the presence or absence of cancer.

	Depressive Symptoms (EURO-D 12 Score)					
	Total		Male		Female	
	Mean (SD)	*p*	Mean (SD)	*p*	Mean (SD)	*p*
With Cancer	3.03 (2.31)	<0.001	2.66 (2.2)	<0.001	3.39 (2.24)	<0.001
Without Cancer	2.28 (2.14)		1.86 (1.9)		2.59 (2.40)	

Abbreviations: SD, standard deviation.

**Table 4 healthcare-13-00715-t004:** Moderation analysis of grip strength for the relationship between cancer and depressive symptomatology, adjusted for age and sex (total sample).

	Depressive Symptoms (EURO-D 12 Score)					
	Total Sample		Male		Female	
	B	95% CI	B	95% CI	B	95% CI
Cancer (X)	1.09	0.81, 1.37	1.65	1.13, 2.17	1.17	0.66, 1.68
Grip Strength (W)	−0.04	−0.04, −0.04	−0.02	−0.03, −0.02	−0.05	−0.06, −0.05
Cancer Grip Strength	−0.01	−0.02, −0.01	−0.03	−0.04, −0.01	−0.02	−0.04, 0.00

Abbreviations: CI, confidence interval. Note: Regression coefficients (B) are unstandardized.

## Data Availability

The data are freely accessible. The data can be accessed through the SHARE project website—https://share-eric.eu/ (21 September 2024).

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
