# Peer review of "Moderating Effect of Muscular Strength in the Association Between Cancer and Depressive Symptomatology"

_healthcare, 2025, doi:10.3390/healthcare13070715_

Round 1

Reviewer 1 Report

Comments and Suggestions for Authors

Moderating effect of muscular strength in the association between cancer and depressive symptomatology

 – Peer Review

I appreciate the opportunity to review this manuscript. This manuscript titled “Moderating effect of muscular strength in the association between cancer and depressive symptomatology” used a large publicly available European dataset to assess associations between a diagnosis of cancer and depressive symptoms measured using the EURO-D 12-item scale. Further, the authors assessed if objectively measured grip strength modified the relationship between cancer and depression. Overall, the authors should be commended for the conceptualization of the paper as this topic is pertinent and grip strength (or muscular strength) is a modifiable risk factor that could be targeted with lifestyle intervention. Moreover, the authors stratified their analysis by sex, which adds much-needed female evidence to the literature.

However, there are substantial areas for improvement. Notably, although the authors do point out their effect sizes were quite small and that there was a lack of control for multiple confounders, this point should not be understated. Their main modification effects were bordering on the null, leaving one to wonder if the addition of one more relevant confounders or operationalization of the existing confounder (self-reported physical activity) or outcome (cut-point used for the EURO-D12) would have nullified the result. A sensitivity analysis might be prudent.

Clarification to the methods and the provision of rationale for their decisions are needed. For example:

Line 112-115: “GS variable contained the maximum value of the GS measurement of both hands. Valid measurements were values of two in one hand that differed by less than 20 kg. GS measurements with values equal to 0 kg or higher than 100 kg were excluded, as well as if GS was only measured once in one hand.”

First, a variation of 20kg seems large, in fact it is close to the mean in the female strata. Is there justification for this not being a lower value? Second, it is assumed that this indicates the maximum value of the left OR the right hand and not the sum of BOTH of the left and right hand. Is this a correct assumption? Based on the reported mean for the sexes for grip strength and the participants’ corresponding age, this group obtained quite the high relative grip strength. For example, using Canadian norms for healthy females aged 70-74, the 75th percentile is 25.3kg, which is very close to the estimates for this current paper, which are participants diagnosed with cancer. Using this line of reasoning, another passage (Line 205-207, “However, the finding may serve as a strategy capable of alerting and assisting professionals about the presence of depressive symptoms in the presence of values below 55.3 kg for men and 39.4 kg for women.”) would warrant clarification, as these values are well in the upper 90s for percentile comparison.

Other minor comments include:

Line 335-36: “Global statistics indicate that by 2022 there will be approximately 20 million new cases of cancer, followed by 10 million deaths [1].” – The year 2022 is now past. This sentence needs revision.

Line 50 – To strengthen the argument about the relationship between cancer and low muscle strength, one could introduce evidence of cancer cachexia.

Line 62-63: “was responsible for an increased HR of 1.01 for cancer [14]” – Was it causal or was this just an association? What was the confidence interval for the HR? If the point estimate is 1.01, the confidence interval may be null.

Table 3 – Need to note the difference between male and female. As is, it seems as though the only comparisons being made are within cancer diagnosis, although the text makes reference to difference in sex, line 160-162 “Regardless of the presence or absence of a cancer diagnosis, women consistently reported higher average scores of depressive symptoms than men.”

Table 4 – Need to indicate that these are adjusted estimates. Further, are these standardized or unstandardized betas? It is assumed they are unstandardized based on the symbol, but clarification would be helpful.

Overall

Missing multiple punctuation points throughout.

Many sentences could use more specific wording. For example, line 181-182: “In line with previous studies, depressive symptoms were prevalent in women [23, 24].” Should this sentence have the qualifier ‘more prevalent compared to males’?

If possible, write out muscle strength instead of MS, and grip strength instead of GS. Further, although grip strength is a proxy for muscle strength, it would be prudent to use grip strength in wording to be more specific – where applicable.

It is great to see estimates based on sex (male and female). However, it would be prudent to use male and female throughout and avoid man and women, which would be best suited for gender. Further, as the authors stratify on sex, it would strengthen their argument if sex was introduced in the introduction. 

Comments on the Quality of English Language

Included above

Author Response

Comment: Clarification to the methods and the provision of rationale for their decisions are needed. For example:

Line 112-115: “GS variable contained the maximum value of the GS measurement of both hands. Valid measurements were values of two in one hand that differed by less than 20 kg. GS measurements with values equal to 0 kg or higher than 100 kg were excluded, as well as if GS was only measured once in one hand.”

First, a variation of 20kg seems large, in fact it is close to the mean in the female strata. Is there justification for this not being a lower value?

Response: Thank you for your comment. Data from SHARE have shown that there are significant variations in grip strength among European Union countries. For example, countries such as Sweden and Denmark exhibit, on average, higher grip strength values compared to Southern European countries like Portugal and Italy. The authors of SHARE considered 20 kg to be an appropriate value to, on the one hand, establish a limit that helps prevent measurement errors or incorrect records that could distort data analysis, and, on the other hand, facilitate comparisons between populations with significant differences in physical health, mobility, functional capacity, and mortality.

Comment: Second, it is assumed that this indicates the maximum value of the left OR the right hand and not the sum of BOTH of the left and right hand. Is this a correct assumption?

Response: Thank you for your comment. Yes, that is a correct assumption.

Comment: Based on the reported mean for the sexes for grip strength and the participants’ corresponding age, this group obtained quite the high relative grip strength. For example, using Canadian norms for healthy females aged 70-74, the 75th percentile is 25.3kg, which is very close to the estimates for this current paper, which are participants diagnosed with cancer. Using this line of reasoning, another passage (Line 205-207, “However, the finding may serve as a strategy capable of alerting and assisting professionals about the presence of depressive symptoms in the presence of values below 55.3 kg for men and 39.4 kg for women.”) would warrant clarification, as these values are well in the upper 90s for percentile comparison.

Response: Thank you for your suggestion. The Johnson-Neyman test is a statistical approach used to interpret the results of a moderation analysis, especially when there is a significant interaction between the moderator variable and the independent variable. This test allows for the identification of thresholds and regions of significance, that is, the values of the moderator variable for which the relationship between chronic diseases and depression is statistically significant.

The regions of significance define the ranges of muscle strength values in which the relationship between chronic diseases and depression is statistically relevant. This provides a more detailed understanding of the role of muscle strength as a moderator. If, for example, the results indicate that the relationship is significant only in individuals with low muscle strength, this suggests that interventions focused on muscle strengthening could have a meaningful impact on reducing these individuals' psychological vulnerability to chronic diseases.

When analyzing the number of participants in the sample who fall below these cutoff values for grip strength (GS), we find that they correspond to 90%–95%. In practical terms, this means that this moderation effect applies to almost all individuals.

Although further research is needed, the fact that this moderation effect is observed across the entire strength spectrum, for both men and women, suggests that an individual diagnosed with sarcopenia—who, despite participating in a muscle-strengthening program, has not yet reached the muscle strength cutoff (27 kg for men and 16 kg for women)—is already experiencing a mitigating effect on depressive symptoms. If intervention programs aim to integrate strategies for promoting both physical and mental health, this finding appears to be of significant importance.

Comment: Line 335-36: “Global statistics indicate that by 2022 there will be approximately 20 million new cases of cancer, followed by 10 million deaths [1].” – The year 2022 is now past. This sentence needs revision.

Response: Thank you for your comment. We have improved the sentence accordingly.

Comment: Line 50 – To strengthen the argument about the relationship between cancer and low muscle strength, one could introduce evidence of cancer cachexia.

Response: Thank you for your comment. We have improved the manuscript accordingly.

Comment: Line 62-63: “was responsible for an increased HR of 1.01 for cancer [14]” – Was it causal or was this just an association? What was the confidence interval for the HR? If the point estimate is 1.01, the confidence interval may be null.

Response: Thank you. We have improved the manuscript accordingly.

Comment: Table 3 – Need to note the difference between male and female. As is, it seems as though the only comparisons being made are within cancer diagnosis, although the text makes reference to difference in sex, line 160-162 “Regardless of the presence or absence of a cancer diagnosis, women consistently reported higher average scores of depressive symptoms than men.”

Response: Thank you for your comment. The p-value is presented for both men and women and indicates the statistically significant difference in depressive symptom scores between those diagnosed with cancer and those without.

Comment: Table 4 – Need to indicate that these are adjusted estimates. Further, are these standardized or unstandardized betas? It is assumed they are unstandardized based on the symbol, but clarification would be helpful.

Response: Thank you for your comment. We changed the title of table 4 to indicate that these are adjusted estimates for age and sex. These are unstandardized betas.

Comment: Many sentences could use more specific wording. For example, line 181-182: “In line with previous studies, depressive symptoms were prevalent in women [23, 24].” Should this sentence have the qualifier ‘more prevalent compared to males?

Response: Thank you for your suggestion. We have improved the sentence accordingly.

Comment: If possible, write out muscle strength instead of MS, and grip strength instead of GS. Further, although grip strength is a proxy for muscle strength, it would be prudent to use grip strength in wording to be more specific – where applicable.

Response: Thank you for pointing this out. We wrote out muscle strength instead of MS, and grip strength instead of GS across all manuscript.

Comment: It is great to see estimates based on sex (male and female). However, it would be prudent to use male and female throughout and avoid man and women, which would be best suited for gender. Further, as the authors stratify on sex, it would strengthen their argument if sex was introduced in the introduction. 

Response: Thank you for your suggestions. We have improved the manuscript accordingly.

Reviewer 2 Report

Comments and Suggestions for Authors

This study explores the moderating role of muscle strength between cancer and depressive symptoms. The study utilizes data from a large-scale survey and provides a clear analytical logic. I offer a few minor suggestions for the authors' consideration.

The third paragraph of the introduction needs improvement. It would be more appropriate to directly start with the comorbidity of cancer and depression.

It would be best to provide a detailed explanation in the introduction of why grip strength was chosen as the indicator for strength.

The authors used a large sample from a large database, which enhances the credibility of the study.

The authors treat strength as a moderating factor, so the introduction should include some relevant evidence. Currently, the evidence mainly focuses on the association between them two, which is a different perspective from a moderating effect.

I suggest clarifying the process for participant inclusion. Were all the samples obtained from the database used in the analysis? As far as I know, some samples in such databases are often excluded from analysis due to various reasons (e.g., data missing).

The authors included covariates and created a DAG diagram, which made the analysis very clear. However, I recommend supplementing the selection of covariates with some literature, as correctly specifying covariates is often evidence-based. This will increase our confidence in the results.

According to the STROBE guidelines, it would be beneficial to include sensitivity analyses to provide additional insights and double check the robustness of the evidence.

The discussion should include a limitations section, stating areas where the study could be improved and exploring the internal and external validity of the results.

Author Response

Comment: The third paragraph of the introduction needs improvement. It would be more appropriate to directly start with the comorbidity of cancer and depression.

Response: Thank you for pointing this out. We have improved the third paragraph to better explain the comorbidity of cancer and depression.

Comment: It would be best to provide a detailed explanation in the introduction of why grip strength was chosen as the indicator for strength.

Response: Thank you for your suggestion. We added information in the last paragraph of the introduction of why grip strength was chosen as the indicator for strength.

Comment: The authors treat strength as a moderating factor, so the introduction should include some relevant evidence. Currently, the evidence mainly focuses on the association between them two, which is a different perspective from a moderating effect.

Response: Thank you for your comment. To the best of our knowledge, there is no evidence in the scientific literature that examines the moderating effect of muscle strength on the relationship between cancer and depression. There are studies that analyze the association between these variables, using simpler or more complex models with varying numbers of variables involved, but they do not conduct a moderation analysis; rather, they assess the association between these variables. As such, it is not possible to include relevant evidence in the introduction regarding the moderating effect of muscular strength on these variables.

Comment: I suggest clarifying the process for participant inclusion. Were all the samples obtained from the database used in the analysis? As far as I know, some samples in such databases are often excluded from analysis due to various reasons (e.g., data missing).

Response: Thank you. We added information about participant inclusion in the “Materials and Methods” section.

Comment: According to the STROBE guidelines, it would be beneficial to include sensitivity analyses to provide additional insights and double check the robustness of the evidence.

Response: Thank you for your suggestions. We recognize the value of sensitivity analyses in observational studies. However, for this specific type of study (cross-sectional), sensitivity analyses are not typically required, as our primary objective was to analyze how muscular strength moderates the relationship between cancer and depressive symptomatology among older European adults. Additionally, in our study, we employed a moderation analyses, which inherently accounts for potential biases and variability in the data. Given the robustness of our primary analytical approach and the consistency of our findings, we believe additional sensitivity analyses would not significantly alter the conclusions.

Comment: The discussion should include a limitations section, stating areas where the study could be improved and exploring the internal and external validity of the results.

Response: Thank you for your comment. We included a limitations section in the discussion.

Reviewer 3 Report

Comments and Suggestions for Authors

One of the strengths of this article is conducting interdisciplinary research( sport medicine, oncology and psychiatry) and the article has the idea of innovation.

In terms of psychiatry, the diagnosis of depression is not clear. The diagnosis should be based on international criteria such as DSM or ICD, and the questionnaires should be standard( such as  Beck or Hamilton questionnaires).

Depression is a biopsychosocial disorder, and the direct relationship between increasing muscle mass and treating depression is simplistic.  

Author Response

Comment: In terms of psychiatry, the diagnosis of depression is not clear. The diagnosis should be based on international criteria such as DSM or ICD, and the questionnaires should be standard (such as Beck or Hamilton questionnaires).

Response: Thank you for pointing this out. The literature reports that conclusions regarding the incidence of mental health disorders among older adults are often inconsistent. This inconsistency can be attributed not only to cultural differences but also to a lack of measurement equivalence across different groups of older adults. In other words, different assessment instruments are used, and even when the same instrument is consistently applied, its validity may be questioned when used with culturally diverse samples. Although the participants are all Europeans, cultural differences between countries are significant, especially when it comes to constructs related to mental health. To allow for a meaningful interpretation of similarities and differences in scale scores in cross-country comparative studies, the scale must measure a single construct and be equivalent across different country samples. Validation studies of the EURO-D have demonstrated its internal consistency and validity in various European contexts. The scale has good psychometric properties, making it useful for international comparisons of depressive symptoms in older adults.

Comment: Depression is a biopsychosocial disorder, and the direct relationship between increasing muscle mass and treating depression is simplistic. 

Response: Thank you for your comment. While recognizing the complexity of these phenomena, the literature highlights the need to approach this complexity not as an endpoint, but as a challenge for reflection. This involves alternating between simpler models that deconstruct the phenomenon and more complex models that represent the system of associations as a whole. We believe that our model aligns with this approach to deconstruction and, as such, allows us to better isolate the relationships between the variables of interest (muscle strength, chronic diseases, and depression).

Round 2

Reviewer 1 Report

Comments and Suggestions for Authors

Moderating effect of muscular strength in the association between cancer and depressive symptomatology – Revision 1.

 – Peer Review

I want to thank the authors for addressing many of my concerns.

A few additional comments and clarifications are needed.

Response to old comments

Old comment: /*Line 112-115: “GS variable contained the maximum value of the GS measurement of both hands. Valid measurements were values of two in one hand that differed by less than 20 kg. GS measurements with values equal to 0 kg or higher than 100 kg were excluded, as well as if GS was only measured once in one hand.”

First, a variation of 20kg seems large; in fact, it is close to the mean in the female strata. Is there justification for this not being a lower value?*/

Your response to my comment is noted, and understandably, grip strength varies by region and population. However, my comment was aimed at the individual. Having individual measurements vary by up to 20kg is a large intra-individual variation. If this is how the SHARE research team has deemed their data, I understand that you cannot change this; however, it should be noted in the limitations of the study.

Old comment: /*Second, it is assumed that this indicates the maximum value of the left OR the right hand and not the sum of BOTH of the left and right hand. Is this a correct assumption?*/

Thank you for the confirmation; however, please clarify this in the text. Currently, it reads “Grip strength variable contained the maximum value of the grip strength GS measurement of both hands.” Which is still confusing.

Old comment: /*Table 3 – Need to note the difference between male and female. As is, it seems as though the only comparisons being made are within cancer diagnosis, although the text makes reference to difference in sex, line 160-162 “Regardless of the presence or absence of a cancer diagnosis, women consistently reported higher average scores of depressive symptoms than men.”*/

Thank you for your response. When indicating that a score is higher between two groups, it is often thought in a statistical manner, as all parameters come with variation. Therefore, although the point estimates for the females are higher than males, this does not mean they are different on average when variability is considered. This would be assessed formally as you have done with cancer and without cancer.

Old comment /*Table 4 – Need to indicate that these are adjusted estimates. Further, are these standardized or unstandardized betas? It is assumed they are unstandardized based on the symbol, but clarification would be helpful.*/

Thank you for your reply. Adjustment for age would apply to all three columns (i.e., total, male, and female); however, adjustment for sex would only apply to the total. Further, although you clarified that the beta is unstandardized to me, make this clear in the text. Remember, all tables and figures should be standalone.

New comment

Line 69-71: “A valid and reliable indicator of total muscular strength and the condition of one's physical and mental well-being is grip strength, a muscle strength metric that is correlated with levels of physical activity [15]”

This statement is not correct. It is fair to say that grip strength is a valid and reliable measure of total muscle strength, and I have made such claims in the past. However, it is more correct to indicate that grip strength is a valid and reliable indicator of upper body strength and a proxy for total muscular strength. Further, grip strength is not a valid and reliable measure for mental well-being; it is merely associated with it.

Comments on the Quality of English Language

Another detailed proofread is recommended.

Author Response

Comment: Old comment: /*Line 112-115: “GS variable contained the maximum value of the GS measurement of both hands. Valid measurements were values of two in one hand that differed by less than 20 kg. GS measurements with values equal to 0 kg or higher than 100 kg were excluded, as well as if GS was only measured once in one hand.”

First, a variation of 20kg seems large; in fact, it is close to the mean in the female strata. Is there justification for this not being a lower value?*/

Your response to my comment is noted, and understandably, grip strength varies by region and population. However, my comment was aimed at the individual. Having individual measurements vary by up to 20kg is a large intra-individual variation. If this is how the SHARE research team has deemed their data, I understand that you cannot change this; however, it should be noted in the limitations of the study.

Response: Thank you for your suggestion. Indeed, this is how the SHARE research team has deemed their data. In light of this, we enhanced the limitations section.

Comment: Old comment: /*Second, it is assumed that this indicates the maximum value of the left OR the right hand and not the sum of BOTH of the left and right hand. Is this a correct assumption?*/

Thank you for the confirmation; however, please clarify this in the text. Currently, it reads “Grip strength variable contained the maximum value of the grip strength GS measurement of both hands.” Which is still confusing.

Response: I appreciate you bringing this up. We adedd that information to the "measures" sub-section.

Comment: Old comment: /*Table 3 – Need to note the difference between male and female. As is, it seems as though the only comparisons being made are within cancer diagnosis, although the text makes reference to difference in sex, line 160-162 “Regardless of the presence or absence of a cancer diagnosis, women consistently reported higher average scores of depressive symptoms than men.”*/

Thank you for your response. When indicating that a score is higher between two groups, it is often thought in a statistical manner, as all parameters come with variation. Therefore, although the point estimates for the females are higher than males, this does not mean they are different on average when variability is considered. This would be assessed formally as you have done with cancer and without cancer.

Response: Thank you for your comment. When we indicate that a score is higher between two groups, we report that based on a statistical inference method (in this case, t-test for independent samples), which means that variability is considered.

Comment: Old comment /*Table 4 – Need to indicate that these are adjusted estimates. Further, are these standardized or unstandardized betas? It is assumed they are unstandardized based on the symbol, but clarification would be helpful.*/

Thank you for your reply. Adjustment for age would apply to all three columns (i.e., total, male, and female); however, adjustment for sex would only apply to the total. Further, although you clarified that the beta is unstandardized to me, make this clear in the text. Remember, all tables and figures should be standalone.

Response: Thank you for your suggestion. We improved the table accordingly.

Comment: Line 69-71: “A valid and reliable indicator of total muscular strength and the condition of one's physical and mental well-being is grip strength, a muscle strength metric that is correlated with levels of physical activity [15]”

This statement is not correct. It is fair to say that grip strength is a valid and reliable measure of total muscle strength, and I have made such claims in the past. However, it is more correct to indicate that grip strength is a valid and reliable indicator of upper body strength and a proxy for total muscular strength. Further, grip strength is not a valid and reliable measure for mental well-being; it is merely associated with it.

Response: Thank you for your suggestion. We improved the statement accordingly.